

# Quantifying SAGE II (1984-2005) and SAGE III/ISS (2017-2021) observations of smoke in the stratosphere

Larry W. Thomason[1] and Travis Knepp[1]

[1]NASA Langley Research Center, Hampton, VA 23681 USA

*Correspondence to*: Larry W. Thomason (l.w.thomason@nasa.gov)

**Abstract.** Using a common analysis approach for data sets produced by the Stratospheric Aerosol and Gas Experiment instruments SAGE II and SAGE III/ISS, we identify 13 likely smoke events based on enhancements in aerosol extinction coefficient. Nine of these are sufficiently large compared to ambient aerosol levels to compute mean mid-

latitude 1020-nm optical depth enhancements that range from 0.0005 to 0.011. We also note that, for large events, the 525 to 1020 nm aerosol extinction coefficient ratio asymptotes at high extinction coefficient to values between 2 and 3 suggesting that the aerosol is relatively small (<0.3 μm) and relatively consistent from event to event. Most of these events are primarily confined to the lower stratosphere and rarely can be observed above 20 km.  We also infer an increase in the frequency of smoke events between the SAGE II (1984-1991, 1996-2005) and SAGE III/ISS (2017-

present) periods by almost a factor of two and also note that the two largest events occur in the latter data set. However, given the low frequencies overall, we are not confident that the differences can be attributed to changes between the two periods. We also attempt to disentangle the mixing of aerosol in the northern hemisphere summer of 1991 from a pyrocumulus event (Baie-Comeau, Quebec) and Mt. Pinatubo and conclude that, while there is evidence for smoke in the lower stratosphere, virtually all of the enhanced aerosol observations in the northern mid-latitudes in the summer

of 1991 are associated with the Mt. Pinatubo eruption.

## 1 Introduction

The strongest factors to modulate stratospheric aerosol levels are among some of the most spectacular processes that our planet has to offer.  Chief among these are volcanic eruptions like the 1991 Mt. Pinatubo eruption which virtually instantly changed the stratospheric aerosol loading by a factor approaching 100. Other events like the

recent Hunga Tonga eruption, while considerably smaller than Mt. Pinatubo's eruption, also strongly modulate stratospheric aerosol levels (Khaykin et al., 2020). While volcanism is the primary driving factor behind long-term aerosol levels and their variability, other factors are also relevant. The non-volcanic background, while perhaps not the definition of spectacular, is mostly driven by the emission of OCS and other sulfur bearing gas species at the sea or land surface that eventually find their way to the stratosphere and modified into sulfuric acid aerosol (Kremser et

al., 2016). The role of organic aerosol particularly in the lower tropical stratosphere is known but not fully quantified (Murphy et al., 2014).  It has also become more recognized that stratospheric aerosol levels particularly in the extended relatively clean period of the last few decades can be significantly modulated by forest and brushfires (Fromm et al., 2010) through (spectacular) cumulus flammagenitus (https://cloudatlas.wmo.int/en/flammagenitus.html) that are more commonly called pyrocumulonimbus or



pyrocumulus), pyrotornadogenesis, and similar processes. In fact, the intense Australian brushfires in the austral
      summer of 2019/2020 had an impact on stratospheric aerosol levels on par with the moderate Raikoke eruption of
      2019 and clearly enhanced stratospheric aerosol extinction coefficient in the southern hemisphere for well over a
      year (Kloss et al., 2021).  The two largest smoke intrusions into the stratosphere are these Australian fires and a
      pyrocumuluonimbus event associated with forest fires in British Columbia Canada in August 2017. Both of these

events have occurred relatively recently, and this raises the possibility that the impact of intense fire events on the
      stratosphere has changed or is changing possibly as a part of processes associated with on-going climate change
      (Canadell et al., 2021). Herein, we will look at space-based observations of smoke events that span from 1984
      through the present. Each event will be characterized based on the observations provided by the instruments rather
      than by indirectly inferred parameters. Based on these observations, we will discuss differences observed across this

period and ability to infer change using these data. Whatever the source, we refer to aerosol associated with fire-
      related processes as 'smoke' without specific assumptions regarding composition recognizing that such aerosol are
      likely composed of black and/or brown carbon plus additional materials produced by biomass consumption in a
      potentially complicated composite aerosol.

      Since the late 1970s, a number of space-based instruments have made stratospheric aerosol observations (e.g.,

Kremser et al., 2016). These instruments make their measurements using diverse methodologies including solar
      occultation, limb-scattering, and nadir-viewing lidar. Prior to 2002, virtually all of the space-based observations of
      stratospheric aerosol are based on solar occultation. Among these, the Stratospheric Aerosol and Gas Experiment, or
      SAGE, series of instruments have formed a key element of the long-term stratospheric aerosol record. There was a
      break in the solar occultation record following the end of the SAGE II record in 2005 but resumed in 2017 with the

SAGE III mission aboard the International Space Station (ISS). After 2002, the Optical Spectrograph and Infra-Red
      Imaging System (OSIRIS) instrument (Rieger et al., 2019) begins a very long, and continuing, record of higher
      frequency limb-scattering observations and, after 2012, by similar observations by the Ozone Mapping and Profiler
      Suite (OMPS) (Taha et al., 2021). Nadir-viewing lidar observations of stratospheric aerosol began in 2006 with the
      Cloud-Aerosol Lidar with Orthogonal Polarization (CALIOP) instrument apart of the Cloud-Aerosol Lidar and

Infrared Pathfinder Satellite Observation (CALIPSO) (Tackett et al, 2018) that continue to the present. Other space-
      based measurements like the Halogen Occultation Experiment (HALOE), SCanning Imaging Absorption
      spectroMeter for Atmospheric CHartographY (SCIAMACHY), and others can potentially provide relevant
      contributions to understanding smoke in the stratosphere.

      We focus on the records from the Stratospheric Aerosol and Gas Experiment (SAGE II), which flew aboard the

Earth Radiation Budget Satellite (ERBS), spanning 1984 to 2005, and SAGE III/ISS with data since 2017. We focus
      on these records since they are very compatible data sets that produce aerosol extinction coefficient profiles at
      essentially the same wavelengths using the same measurement approach and very similar processing algorithms to
      produce near global observations. Issues related to uncertainties in characterizing the scattering processes add
      potential systematic bias to the records from OSIRIS, OMPS and CALIPSO and make intercomparisons of these

records with SAGE and each other more difficult (Kovilakam et al., 2023; Bourassa et al., 2023) and inclusion of



these data sets is deferred. The SAGE instruments are key sources of data to the Global Space-based Stratospheric Aerosol Climatology (GloSSAC) and a key outcome of this study is to infer the impact of smoke on this climate data record. SAGE II has been the primary data source for GloSSAC and its predecessors (e.g., ASAP) and remains the governing data set against which other contributors are measured. The consistency between SAGE II and SAGE

III/ISS can only be indirectly inferred (Kovilakam et al., 2020), however, based on GloSSAC testing, the two data sets are considered to be highly consistent and are treated as such herein.  Given this measurement similarity, we focus on data from these two instruments in this study.  The goal of this discussion is twofold. One is to assess the frequency, magnitude, and persistence of smoke events in these two data sets and how they impact the GloSSAC climatology. Secondly, to the degree possible, we will infer the differences in the smoke events for the observational

times periods covered by SAGE II and SAGE III/ISS. When discussing the observations in general as opposed to those by a specific instrument, we refer to 'SAGE' observations for simplicity.

We identify likely smoke events in both data sets as spatial/temporal clusters of positive outliers from the broad family of aerosol extinction coefficient observations as a function of time and altitude. While outliers are a part of both records, we do not attempt to associate every isolated positive outlier with its source since we are primarily

interested in events which significantly impact the stratospheric aerosol record at GloSSAC times scales (monthly). While small volcanic and smoke events produce similar perturbations in a single wavelength record, we also demonstrate that for the smoke events discussed below, it is straightforward to distinguish between episodic clusters associated with smoke events and volcanic events in these data sets based on their spectral characteristics.  All of likely smoke cluster events in these data sets can be tied to known large fire events. While we make no effort to infer

the composition of these perturbations, the measured optical properties found for these smoke-derived perturbations are generally consistent with each other and not fresh small to moderate volcanic events. We will also discuss in some detail an outlier event from the summer of 1991 that has been associated with both a volcanic source (Thomason et al, 1992) and a pyrocumulus event (Fromm et al., 2010).  In some cases, while the presence of smoke in the lower most stratosphere may be inferred from the SAGE observations, not all are considered in depth. Some

of these occur in only a handful of observations such as the Conibear Lake fire of 2003 that would be difficult to conclude anything about their impact which ultimately must be small. Others occur at altitudes essentially at the tropopause, particularly the Yellowstone fires of 1988, and we are not able to perform a meaningful stratospheric analysis. We note these events but otherwise do not discuss them.

## 2 Differentiating between outlier events and clouds

Figure 1 shows the 1020-nm extinction coefficient for all SAGE II measurements between 20 and 60N after 1 January 1997 at 11 km with all data above the MERRA-derived tropopause (a), the same data but 'cloud-cleared' using the GloSSAC process (b), the same data as (a) above the tropopause+0.5 km (c), above the tropopause +1.0 km (d), above the tropopause +1.5 km (e) and above the tropopause +2.0 km (e). Given the discrete altitude grid of the measurements, data in frames 1a and 1b are between 0 and 0.5 km above the tropopause and frames 1c through

1f are similarly affected.  The frequency of observations is driven by the sampling characteristic of solar occultation instruments in mid-inclination orbits like both SAGE II and SAGE III/ISS. As can be readily seen in the figure,



following an instrument fault in August 2000, there is a 50% reduction in data taken through the end of the SAGE II mission in August 2005.

In general, all frames in Figure 1 show a relatively consistent pattern throughout the period with an annual cycle with a maximum in summer. However, for the most inclusive frames such as 1a, there is a patina of positive outliers throughout the record. Most of these data points are also associated with decreases in the 525 to 1020-nm extinction coefficient that suggest an increase in particle size of optically effective particles assuming non-absorbing aerosol such as sulfuric acid/water mixtures. In the absence of information suggesting otherwise, high extinction/low ratio data points are most often interpreted as 'cloud' or aerosol/cloud mixtures (e.g., Thomason and Vernier, 2013). This

observation forms the basis for essentially all mechanisms to identify and remove the effects of clouds on the aerosol observations including for such data sets as GloSSAC (Kovilakam et al., 2023). Despite numerous positive outliers in Figure 1a, a cluster of enhanced aerosol extinction coefficient in mid-1998 is clearly seen in this data set in the time frame where material associated with the Norman Wells (NT, Canada) pyrocumulus on 3-4 August 1998 has previously been noted in SAGE II data (Fromm et al., 2000). While the individual observations are similar to

aerosol/cloud mixtures routinely observed by SAGE instruments, the extinction coefficient enhancement, while variable, is observed with a high frequency in a narrow temporal window that is inconsistent with the relatively isolated way cloud-affected measurements appear in the rest of this figure and SAGE data in general. Isolating this, and other similar clusters is a key part of this analysis.

The data in Figure 1a is restricted to observations above the tropopause to minimize the impact of cloud presence.

However, the tropopause height is from MERRA for SAGE II and subject to uncertainties as much as the SAGE II observations. In addition, SAGE II has a finite vertical field of view (0.5 km) and that along with the data accumulation process used in data production further produce a vertical resolution of about 1 km. As a result, it is not surprising that the presence of cloud is occasionally inferred for observations that nominally represent the lower most stratosphere. In many applications such as GloSSAC, removing the effects of cloud presence is a critical part

of their analysis. The GloSSAC aerosol/cloud mixture identification method (Thomason et al., 2018) is based on both the magnitude of extinction coefficient (at 1020 nm) and the 525 to 1020-nm extinction coefficient ratio. Using this procedure, Figure 1b shows the same collection of 1020-nm extinction coefficient observations as in Figure 1a with all identified cloud/aerosol mixtures observations removed. As can be seen, this process does a reasonable job at removing most, but not all, isolated enhanced aerosol extinction coefficient values in this record. However, we

also note that the performance around the elevated extinction coefficient cluster in mid-1998 is concerning as that cluster, as depicted in Figure 1a, clearly extends to an extinction above 0.003 km$^{-1}$ but is truncated by the cloud-clearing process for extinction coefficient values at about 0.002 km$^{-1}$ as shown in Figure 1b. As we will show below, this truncation occurs because these data points also show a decrease in aerosol extinction coefficient ratio and thus their behavior mimics behavior normally associated with aerosol/cloud mixtures and is identified as such by the

cloud clearing process. This truncation of the enhanced aerosol cluster suggests that GloSSAC currently does not currently do a good job depicting this event in the data set and some modification of the cloud detection scheme is warranted to properly account for these perturbations.



Since we are focusing on identifying outlier events in the lower stratosphere, clearly employing a cloud identification process that removes a significant fraction of the target events is not appropriate and an alternative

method to retain this sort of enhancement that also limits the influence of clouds on the data analysis is required. We use a straightforward tropopause-based altitude filter that eliminates observations close to but above the reported tropopause altitude. In Figures 1c to 1f, each step limiting data's proximity to the tropopause decreases the overall amount of data but clearly reduces the frequency of the aerosol/cloud mixture patina with each step through at least +1.5 km (Figure 1e). In addition, while there is some impact on the number of data points in the enhanced cluster in

mid-1998, it is substantially less affected than using the traditional cloud-clearing approach and the highest extinctions remain over 0.003 km$^{-1}$. Furthermore, in Figures 1e and 1f, an additional, but much weaker cluster of enhanced extinction coefficient is now more clearly seen in mid-2001, which will be discussed below, as are some small clusters of enhanced aerosol extinction that may be related to the eruptions of Korovin in May/June 1998, Shishaldin in April 1999 (which is clearer at other altitudes), and Hekla in February 2000 (Pieri et al, 2001). While

tropopause limited filtering works well at all altitudes, a downside of this approach is that inevitably some stratospheric measurements, including outlier measurements, either volcanic or smoke, are eliminated from further consideration. This may impact the inferred frequency of outlier observations and the estimation of the optical impact of all lower stratospheric outlier events due to the proximity of the tropopause. Nonetheless, this is the most straightforward and effective mechanism we have found that minimizes the influence of clouds in the analysis

without strongly impacting the frequency and strength of outlier events. Thus, we use tropopause+1.5 km to filter data used below. With this filter, we find that we have sufficient data to produce a meaningful analysis above between 9.5 and 12 km depending on time of year and the overall sampling frequency provided by the instruments. We note that some instances of isolated high aerosol extinction coefficients remain in the data set even after this filtering. While these observations are interesting, we will not consider these aerosol observations since they occur

at a low frequency and ultimately have only a marginal impact on the data record. We will focus on positive outlier clusters (multiple observations) occur spanning multiple altitude levels.

**3 Identifying and depicting outlier events**

Using data with the tropopause-based altitude filter (TBA), we look at individual years (or two years in some cases) for relatively small, transient outlier events (lasting months as opposed to years), in northern and southern mid-

latitudes. We focus on time periods where potential outlier events appear in analyses like the one shown in Figure 1e (with TBA) which we produced for altitudes below 30 km in northern and southern mid-latitudes in three time periods (1985-1991, 1996-2005, and 2017-2022). Rather than show all of the individual figures, we show the presence of these outliers in a different way that we describe below. While our primary interest is to characterize outlier events associated with pyrocumulonimbus and other smoke intrusions into the stratosphere, we also note

some small volcanic events, and, in one case, an unusual polar stratospheric cloud event whose presence in the SAGE data sets are not generally noted. Conversely, we purposely neglect well-known, larger volcanic events, shown in Table 1, in the SAGE II and III records as not relevant to the current topic. While we have examined the entire SAGE II record for evidence of outliers, we have not done detailed analyses, nor will we further discuss the





period from 1992 to 1995 since the elevated aerosol levels could easily mask the presence of weak outlier events by
volcanism or fires. The British Columbia pyrocumulus of 2017 and Australian wildfires of 2019/2020, which stretch
the definition of small, transient events, are included in our analysis primarily since the intended culmination of this
work is place them in the context of other, smaller smoke events observed by SAGE II and SAGE III using the same
analytic approach.

The data in Figure 1 is for a single altitude (11 km) chosen specifically because the enhancement in northern
hemisphere summer of 1998 is the largest at that altitude.  However, we observed similar enhanced aerosol
extinction coefficients, but with decreasing frequency, in the same time frame as low as 8 km, where TBA filtering
has drastically reduced the available data, and as high as 19 km. Figure 2 shows the distribution of 1020-nm
extinction coefficient observations versus 525 to 1020 nm extinction coefficient ratio (or simply extinction ratio) for
1998 in 12 30-day segments ('months') from Julian days 1-30 to 331-360 for all data passing TBA filtering. The
distributions through Julian day 240, Figure 2 frames (a) to (h), are typical of non-perturbed months with extinction
coefficient inversely correlated with extinction ratio with the maximum extinction and the largest extinction
coefficients, in the densest part of the data scatter, nearly constant or, in this case, slowly decreasing as the Mt.
Pinatubo enhancement continues to ebb. In benign situations, where aerosol is considered to be primarily sulfuric
acid aerosol, extinction ratio is generally inversely correlated with the size of the optically dominant aerosol. In
these sorts of monthly plots, there is generally a strong correlation between lower extinction coefficient values and
increased altitude. Thus, smaller apparent aerosol size occurs with increased altitude.  The effect of the Norman
Wells pyrocumulus makes it presence obvious in Figures 2i and 2j (roughly September and October) with a
considerable change from previous periods as the maximum extinction coefficient values increases from about
$2 \times 10^{-4}$ km$^{-1}$ to $2 \times 10^{-3}$ km$^{-1}$ or about an order of magnitude. In addition, the extinction ratio at the largest extinctions
values drops from a little over 3 to about 2. In the final two periods of 1998, Figures 2k and 2l, distributions have
begun to substantially return to values observed to prior to the perturbation.

To help illustrate the scope and characteristics of the outlier observations, we define two lines based on data in 2
clean periods in the overall analysis period (1-30 and 31-60 in this case). The first line is defined as the 99.5-
percentile of 1020-nm extinction coefficient observations where it exceeds $10^{-4}$ km$^{-1}$ in these clean months. This is
the vertical line shown in all Figure 2 frames. A second line is defined as the median extinction ratio for the same
subset of data and is shown in Figure 2 as the horizontal line between the extinction coefficient line and the
righthand side of the figures. These lines divide the space into three crude zones which roughly correspond to
unenhanced aerosol (to the left of the extinction coefficient line) and, on the righthand side, enhanced aerosol with a
reduced extinction ratio (the lower section and referred to as Type 1) and increased aerosol extinction coefficient
with an increased extinction ratio (the upper section and referred to as Type 2). It is not uncommon in any month for
a handful of observations to occur in either category due to the statistical nature of the categorization process that
occur most often at the fringes of the zones, the incomplete removal of clouds (primarily in Type 1), the presence of
polar stratospheric clouds, and sometimes simply due to questionable quality measurements. The frequency of these
incidental occasions of either type, as will be shown below, is almost always very low and rarely exceeds 5% in any





altitude/month. In situations where stratospheric aerosol levels are trending, usually lower, over the analysis period
       (as can be seen in Figure 2 frames (a) through (h)), the enhancement demarcation can be rather conservative in
       identifying any type of outlier and thus may produce lower Type 1 or 2 frequencies than more generous demarcation
       lines.

       Thomason et al. (2020) showed that small to moderate volcanic events often cause an increase in extinction ratio and
in the context used here those events would be categorized as Type 2 enhancements. Beyond this situation,
       significant numbers of observations in the Type 2 region are not observed. The most common denizens of the type 1
       zone are aerosol/cloud mixtures (e.g., Kent et al., 2003) which for these plots have been mostly removed.  Despite
       TBA filtering, Figure 2i and 2j show substantial numbers of Type 1 enhancements at levels far beyond the frequency
       of such observations in other, non-perturbed periods. Perturbations that move the extinction ratio toward 1 (and
Type 1) are most often associated with cloud presence or intense volcanic events producing very large sulfuric acid
       aerosol such as following the Mt. Pinatubo eruption of 1991. These are primarily scattering particles (as opposed to
       absorbing) at SAGE wavelengths and decreasing extinction ratio is a response to an increase in the size of the most
       optically active aerosol. The asymptotic behavior occurs because, the relative mix between standard aerosol and
       cloud shifts toward cloud, aerosol extinction coefficient increases at both 525 and 1020 such that the extinction ratio
becomes progressively more reflective of cloud properties. Since most ice cloud particles observed by a SAGE
       instrument are optically large, they essentially all have extinction ratios of about 1 and the more cloud-like the
       observation the closer the extinction ratio approaches 1. This is a well-known phenomenon and the basis for
       essentially all cloud detection algorithms for SAGE-like instruments (Thomason and Vernier, 2013; Kovilakam et
       al., 2023). For the Norman Wells event, as can be in Figures 2i and 2j, the extinction ratio asymptotes toward
something closer to 2 and thus the behavior is similar to SAGE ice cloud observations in possessing an asymptotic
       behavior that suggests that the measurements are dominated by an optically uniform aerosol from measurement to
       measurement at high extinctions but to a distinctly not-cloud ratio. Below we will show that this behavior is
       characteristic of smoke events and discuss what this behavior suggests about the observed smoke aerosol further
       below.

Figure 3 shows the temporal evolution of the mean zonal 1020-nm aerosol extinction coefficient (a), the mean zonal
       extinction ratio (b), and Type 1 (c) and Type 2 (d) frequency during 1998 in 10-day segments. The latitudes of
       observations are also shown. In general, we observe an enhancement in aerosol extinction beginning around July 8[th]
       (Julian day 190) below 14 km that spreads to about 18 km beginning in early September (252).  Given that the
       Norman Wells fire occurs on August 4[th] (216), the aerosol enhancement seen prior to that date must arise from a
different source. That source is most likely the June 30[th] eruption of Korovin in the Atka Volcanic Complex (Global
       Volcanism Program, 1998) which produced an ash cloud at 9 km as reported by aircraft. These 'early' aerosol
       enhancements show a low overall frequency of Type 1 events (no more than 20% of observations and only at
       altitudes of 13 km and below). Despite this low frequency the mean extinction ratio reaches values less than 1.6
       suggesting the presence of very large aerosol possibly ash. Beginning in early September, the frequency exceeds
80% at levels below 13 km through early November with at least some Type 1 enhancements as high as 18 km.





During this period, the mean extinction ratio is less than 2.6 in the optically densest part of the layer which is significantly less than the earlier periods. The delay in observations of the Norman Wells aerosol is due to the observational pattern of SAGE II which did not provide northern mid-latitude observations in August 1998. Virtually no observations of Type 2 aerosol are made through the entire year except in the lower most regions with

data (<12 km) where measurement noise and incomplete cloud removal by TBA filtering tends to produce low frequency of occurrences of both aerosol types in all months and years.

Using the analysis in Figure 3a, we can compute a maximum 10-day, zonal magnitude of the perturbation to the stratospheric aerosol optical depth associated with the Norman Well pyrocumulus event and the preceding (probably) Korovin volcanic event. This is straightforward when perturbations are large such as following the

Australian fires of 2019/2020, but it can be difficult when the enhancement is small relative to the background level and for absolute optical depth perturbations magnitude on the order or less than 0.001. The latter difficulty exists because such small variations approach the precision limit of these measurements. In addition, variations in aerosol levels vary seasonally and long-term trends can mask small perturbations and impose an added challenge for inferring an optical depth perturbation for some of the very weak events we discuss. For Norman Wells fire, we find

a maximum 1020-nm optical perturbation of 0.002 occurring with the first observations which is slightly larger than optical depth prior to the event. This enhancement declines to about 0.0005 by the end of the year. For Korovin, we find an optical depth increase of about 0.001 that decreases rapidly after the initial observations as this event appears to have been an extremely short-lived stratospheric feature. From a relative sense both of these perturbations are significant relative to the prior optical depth level (~0.0015), however, the stratospheric aerosol optical depth in

1998 was approaching the lowest ever observed by SAGE II (which occur in 2000). For each event discussed below, we note the maximum estimated optical depth enhancement in Table 2.

In the next section, we apply the above process to the SAGE II (1984-2005) and SAGE III/ISS (2017-2022) observations in the northern and southern mid-latitudes (30 to 60 degrees) where observations are plentiful and, unlike low latitudes, intrusions of smoke into the stratosphere most often occur. For the SAGE III component of the

analysis, we use data from a slightly different wavelength relative to SAGE II (521 nm instead of 525 nm) but we do not distinguish between the extinction ratios computed using SAGE II and SAGE III data considering the small change in wavelength to be of minor importance. From these data, we have identified a total of 18 outlier events which we list in Table 2. Five are identified as non-smoke events and we discuss these briefly either separately below or as part of the discussion of another event when they occur concurrently (as with Norman Wells and

Korovin above). Of the remaining 13, two, as previously noted, are not amenable to further analysis due to their limited impact on the TBA filtered stratosphere (Yellowstone and Conibear Lake). An additional three outlier events identified as originating from fires are shown in Figure 7 but show similarly small enhancements and/or low frequency of observations. These are the Canberra, Australia event of 2003 (Fromm et al, 2008) that had a zonal mean peak 1020-nm optical depth enhancement of about 0.0005, and the Redding California (Carr) fire of 2018

(Lareau et al., 2018) and the McKay Creek fire of 2021 (https://earthobservatory.nasa.gov/images/148530/blazes-rage-in-british-columbia) for which column optical depth enhancements could not be determined. Of the remaining



eight, Norman Wells is discussed above and six are attributed primarily to pyrocumulus events and similar fire-related phenomena and discussed in some detail below. Finally, in a separate section, we discuss the complex period in the summer of 1991 where aerosol from the Mt. Pinatubo eruption and smoke from Baie-Comeau are 290 present in the northern hemisphere lower stratosphere.

In analyzing these events, we employ the typing system categories described above using the two demarcation lines. These are determined based primarily on data from winter months in each hemisphere (days 1 to 60 in the north and 211 to 270 in the south) except some years where this was not possible. In particular, we use Julian days 158 to 210 data in 2017 for the Northern Hemisphere given that day 158 (June 7th) is the start for the SAGE III/ISS record and 295 unaffected by any obvious outlier activity. In the southern hemisphere, we use wintertime data from 2019 to set the demarcation lines since the enhancement from the Australian fire events of 2019/202 persists throughout 2020. We use data from 121 to 180 for the southern hemisphere in 1991 given the massive and persistent effect of the Mt. Pinatubo eruption throughout the second half of 1991. There is considerable variability in the austral winter months in the SAGE II data as both a very clean polar vortex and polar stratospheric clouds are frequently observed. These 300 features produce apparent outliers but are not included as a part of this analysis. Two SAGE II periods are not closely examined. These are 1984 when only 3 months of data are available and is otherwise unremarkable and 1992 through 1995 when background levels from the 1991 Mt. Pinatubo eruption are sufficiently high that small perturbations from smoke or small eruptions could easily be masked by existing aerosol levels.

**4 Application to outlier events**

**4.1 Non-smoke related outliers**

While our interest is primarily in outlier events caused by pyrocumulus and other fire-related sources, we also note a few events that produced clear outliers but are, perhaps, less recognized as being a part of the SAGE II data set. For instance, Pitts et al. (1990) reported on the unusual presence of ice polar stratospheric clouds (PSCs) down to 50N over Europe using SAGE II data from February 1989. While PSCs are commonly observed by SAGE II in the 310 Antarctic mainly in late austral winter and early spring (August and September), they are not common in SAGE II observations in the northern hemisphere and the frequency and latitude of those seen in 1989 are unique in the data set. The effects of these clouds can be seen in Figure 4a where the enhanced aerosol between Julian days 31 and 60 show a general decrease in the extinction ratio suggesting the presence of relatively large particles. A modest zonal increase in 1020-nm aerosol extinction coefficient can be seen around Julian day 50 between 18 and 25 km in Figure 315 4b with a concomitant decrease in extinction ratio, shown in Figure 4c over a similar altitude range. The frequency of these observation is quite low and barely exceed 1% of the observations at any given 10-day period and show essentially no enhancement of Type 2 observations. The low overall frequency isn't surprising given the limited longitudinal range of PSC occurrences in the Arctic and the low likelihood of PSC occurrences in this latitude band in general. The total 1020-nm optical depth anomaly for this event peaks at approximately 0.0005.



Like the signal from the eruption of Korovin, observation of the eruption of Shishaldin in April 1999 is not
        commonly included among volcanic events in SAGE II data set. An ash plume from the April 19th eruption was
        reported up to altitudes between 15 and 20 km with a 63 kt $SO_2$ emission reported below 14 km (Global Volcanism
        Program, 1999). The SAGE II observations shown in Figure 5a for Julian days 121-150 are consistent with these
        observations showing a mix between Type 1 observations, likely the ash, and Type 2 observations which may reflect

new particle formation of sulfuric acid aerosol consistent with observations of other small to moderate volcanic
        events seen in SAGE observations (Thomason et al., 2020). Figure 5b shows the 1020-nm extinction coefficient
        over this period with a modest increase in aerosol extinction below 20 km starting around Julian day 130 after a gap
        in observations. Overall changes in extinction ratio are unremarkable in this period and not shown. On the other
        hand, while frequencies are low, the Type 1 frequency (shown in Figure 5c, exceeds 1% over a broad range of

altitude from the lowest most observations to about 19 km. Type 2 frequencies, shown in Figure 5d, while smaller in
        magnitude than the Type 1 enhancements exceed 10% consistently below 13.5 km with lower frequencies up to
        about 15 km. Both of these features are consistent with other reports regarding the eruption (Global Volcanism
        Program, 1999). The total optical depth anomaly associated with this eruption at 1020 nm is about 0.0008.

        During the airborne SAGE III Ozone Loss and Validation Experiment (SOLVE) in 2000 (completed unfortunately

without SAGE III aboard Meteor 3M), the NASA DC-8 flew through an ash cloud (February 28th) from a recent
        eruption of Hekla (February 26th) at 37000 feet or about 11 km causing some damage to the aircraft (Pieri et al.,
        2001). There was also $SO_2$ emissions of 183 kt at and below 11 km. SAGE II observations shown in Figure 6a for
        Julian days 61-90 show enhancements in both Type 1 and Type 2 categories that persist into April primarily in the
        Type 2 category. The 1020-nm extinction coefficient analysis for 2000 is shown in Figure 6b and shows an increase

primarily confined below 12 km which matches the altitudes where both types of enhancement are also found. The
        extinction ratio analysis shown in Figure 6c shows an intense increase in extinction ratio suggesting large ash
        particles near the altitude where the NASA DC-8 encountered them. Like the previous events, the maximum optical
        depth increase at 1020 nm is small and peaks at about 0.0005. As discussed before, all optical depth perturbations
        below about 0.001 should be considered less robust than larger values given a number of measurement-based and

geophysical factors.

### 4.2 Smoke-related outlier events

        Figure 7 shows the scatter plots, 1020-nm extinction coefficient versus extinction ratio, for 10 smoke events
        including the Norman Wells fire of 1998 as Figure 7c which has been discussed in detail above. These include some
        smoke events whose presence in the SAGE data sets has already been noted: fires located near Circle, Alaska in

1990 (Fromm et al, 2010) in Figure 7b, Chisholm, NT, Canada in 2001 (Rosenfeld et al., 2007; Fromm et al., 2008)
        in Figure 7d, British Columbia in 2017 (Bourassa et al., 2019) in Figure 7f, and the Australian fires of 2019/2020
        (Khaykin et al., 2020) in Figure 7h. In addition, we show observations associated with fires located in the
        Daxing'anling Mountains, China (1987) (Cahoon et al., 1994) in Figure 7a, and the August Complex in California
        (2020) (Keeley and Syphard, 2021) in Figure 7i. These latter events are associated with particular fires based on

spatial and temporal coincidence with known large fire events and the absence of other suitable candidates for the



production of the obvious outlier clusters within the SAGE data set. We also include some minor fire events in this figure including the Canberra fire in 2003 in Figure 7e, the Redding, California (Carr) fire in 2018 as Figure 7g, and McKay Creek, British Columbia, Canada (2021) as Figure 7j.

Taking Figure 7 in total, it is clear that there is substantial variability in both the number of observations (the pervasiveness of outlier observations) and the magnitude of the enhancements with the 2017 British Columbia fire, shown in Figure 7f, and the Australian fires of 2019/2020, shown if Figure 7h, dwarfing the other events in terms of both the number of observations in the enhanced extinction coefficient categories and the magnitude of the enhancements. It is also clear that the Norman Wells fire, now in Figure 7c, is the most pervasive and among the largest magnitude of these events in during the SAGE II period (Figures 7a to 7e). A common feature to these

figures is that virtually all of the enhanced aerosol events fall into the Type 1 category with most events producing close to zero entries into the Type 2 category. As we noted with the Norman Wells smoke event, for none of the events does the extinction ratio appear to asymptote toward 1 though some events are so weak or infrequent that even inferring an asymptotic value exists much less a value for it is not possible. For weaker events this is particularly true since the optical properties of the smoke particles are mixed with those of ambient aerosol rather

than solely represent one composition or the other. While there is often significant scatter in the ratio data, several of the stronger events appear to asymptote toward a value of approximately 2 like the Norman Wells event. These include the Daxing'anling Mountains fire in 1987 (Figure 7a), the Chisholm fire of 2001 (Figure 7d), and the British Columbia fire of 2017 (Figure 7f). In other cases, it appears that the data asymptotes to values can be somewhat larger including the Australian Fire of 2019/2020 that asymptotes to about 3.

It is difficult to infer what changes in observed extinction ratio mean for changes in the underlying aerosol size distribution even when there is little uncertainty regarding composition and refractive index (Thomason et al., 2008). With fire-related events, the interpretation of changes in extinction ratio is more difficult since the complex refractive index may reflect the age of the aerosol, features of the fire itself (what was burning, how hot was it burning) and the process and speed by which it arrived in the stratosphere (Ansmann et al., 2021). The refractive

index may be further modified by interactions with pre-existing aerosol, aerosol precursors like $SO_2$ which may be produced by a fire and other chemical processes (Yu et al., 2019). To our knowledge, *in situ* inferences of the refractive index of smoke-derived aerosol in the stratosphere have not been performed. Existing laboratory measurements of the optical properties of brown carbon span a large range in both real and imaginary components (see the discussion in Knepp et al., 2022). As a result, interpreting the asymptotic values seen in Figure 7 is difficult,

but a few things can be inferred. That an asymptotic value at high extinction coefficient exists suggests that, for measurements that are optically dominated by smoke, the optical properties of the aerosol must be reasonably uniform between measurements in order to produce the relatively tight spread in extinction ratio observed at the highest values of extinction coefficient. This may also suggest that the composition and size distribution of these particles are also reasonably uniform between measurements. If particles are sufficiently large their extinction ratio

must approach one no matter what their complex refractive index is. For scattering aerosol like sulfuric acid/water aerosol and water ice clouds, the 525 to 1020-nm extinction ratio approaches 1 for a particle radius of ~0.5 μm



(Thomason and Vernier, 2013). Figure 8 shows the 525 to 1020-nm extinction ratio for a broad range of single mode log-normal size distributions for both black carbon with refractive index information from Bergstrom et al. (2002) and brown carbon with refractive index information from Sumlin et al. (2018). In these curves, the extinction ratio is strongly dependent on both composition (black vs brown carbon) and size distribution but generally all are approaching a value of 1 by a mode radius of 0.3 μm except the narrowest brown carbon curve where it is closer to 0.4 μm. Since the extinction ratio asymptotes are consistently between 2 and 3, Figure 8 implies that the optically dominant aerosol size is no larger than 0.3 μm and potentially much smaller. It is likely that brown carbon is more prevalent than black carbon in stratospheric smoke (Yu et al. 2019), therefore it is reasonable that the extinction ratios from stratospheric smoke more closely resemble the brown carbon curves of Figure 8 than the black carbon curves. This results in a very coarse determination of particle size where we can conclude that the optically dominant smoke particles were neither very small (e.g., <0.1 μm) nor were they very large (e.g., >0.3 μm) which is in agreement with *in situ* observations (Moore et al., 2021, Katich et al., 2023). Given the uncertainties in the particles' complex refractive index, using SAGE measurements to infer more detailed estimates of the aerosol size distribution or bulk properties like surface area density is far more problematic than when applied to circumstances where sulfuric acid aerosol can be safely assumed. Ultimately, other than concluding that the smoke particles are probably optically dominated by relatively small aerosol, it is not possible to fully separate the effects of composition and size distribution for smoke particles using optical measurements from SAGE-like instruments.

### 4.2.1 Noteworthy events

*Daxing'anling Mountains 1987*

The first outlier event identified as smoke in the SAGE II record is associated with a forest fire (sometimes referred to as the Black Dragon Fire) in the Daxing'anling Mountains, Heilongjiang, China between May 6th and June 2nd, 1987 (Cahoon et al., 1994; Nath and Nath, 2019). Figure 9a shows the 1020-nm extinction coefficient in 1987 in 10-day segments for the 30 to 60N latitude band data in May 1987 (also seen in Figure 7a) show a significant mean change in extinction coefficient confined below 15 km that persists through the end of June and possibly into the late summer. The maximum optical depth enhancement is about 0.002 which decreases substantially by the end of June. Figure 9b shows the frequency of Type 1 enhancements and show that the peak frequency of this enhancement exceeded 40% of all SAGE II measurements around Julian day 160 in the lowest most stratosphere and drops to less than 5% by the end June. Changes in the extinction ratio are unremarkable as the ratio in the primary aerosol layer decreases from ~3 prior to the event to about 2.7 in the optically densest layer immediately afterwards. Additional observations from space of the Daxing'anling Mountains fire were made by the Stratospheric Aerosol Measurement (SAM II). SAM II is the first in the SAGE series of instruments with observations of stratospheric 1000-nm aerosol extinction coefficient between 1978 and 1993 between 60-80° in both hemispheres. Aerosol extinction coefficient at 10 km is shown in Figure 9c with an enhancement shown in beginning around Julian day 150 (June 1st) and lasting to around Julian day 220 (August 8th). During this period, the measurement latitude slowly changes from 65°N to 70°N and provided many observations of this smoke-based aerosol. It is possible, in this case, that SAM II provides



a better view of this event than SAGE II thanks to the fortuitous timing of where the event occurred and was transported in latitude and time relative to where SAM II was making observations. In general, solar occultation measurements in an inclined orbit like SAGE II that provide broad latitude coverage but require several weeks to do

so. As a result, the timing of events relative to measurement latitudes is an additional hurdle to characterizing short-lived phenomena in the stratosphere. Conversely, the Daxing'anling Mountains fire is the only smoke event found in the SAM II record since the timing of observations apparently lacked the requisite serendipity to capture other events such as the Circle Fire of 1990.

*Circle Alaska 1990*

The Circle Alaska fire of 7 July 1990 (Fromm et al., 2010) mean zonal 1020-nm extinction coefficient, shown in Figure 10a, shows an enhanced aerosol extinction starting around Julian day 190 and persisting to around day 300 initially as high as 18 km and with a persistent clear enhancement eventually declining to around 12 km before becoming otherwise undetectable. The peak optical depth enhancement occurs around day 220 with a value on the order of 0.001. There is relatively little change in the mean zonal extinction ratio in through this event with an

average value near 2.8. Conversely figure 7b suggests that the largest enhancements have slightly large extinction ratios (>3) than smaller enhancements. This behavior is reflected in the frequency of Type 1 and 2 observations shown in Figures 10b and 10c. Type 1 observations dominate the early part of the plume occurring up to 18 km and peaking at 14 km at over 40% of observations around Julian day 220. Type 2 events are most common below 13 km and are the dominate enhancement type around Julian day 240 with over 20% of observations at 11.5 km. The

continuous nature of the enhancement shown in Figure 7b (and observed in subsequent months) suggest that the inference of two types shows the limitations of the Type categories as all of this aerosol appears to have the same overall optical character and that, in this case, the asymptotic value is larger than the primary aerosol cluster prior to the event, possibly similar that observed for the Australian fires of 2019/2020, but incompletely expressed due to the small overall enhancement in aerosol extinction.

*Chisholm, Canada 2001*

The zonal mean aerosol extinction for 2001 is shown in Figure 11a with a clear enhancement in aerosol extinction coefficient associated with the Chisholm, Alberta fire of 28 May 2001 confined mostly at and below 17 km. This results in a peak optical depth enhancement of about 0.0018 which is close to that for the Daxing'anling Mountains and Norman Wells fires. These three events form a group of similar sized events as the largest in peak optical depth

in the SAGE II data record. Figure 11b shows that the overall frequency of Type 1 observations in the early summer peaks near 30% of all observations at 13.5 km and exceeds 10% over a broad range of altitudes below 17 km. Despite the low overall frequency of observations of enhanced aerosol coefficient, the aerosol extinction ratio appears to asymptote toward 2 with increasing aerosol extinction coefficient in manner similar to the Norman Wells event. It is possible that the 7 August eruption of Bezymianny (Russia) (Global Volcanism Report, 2001), which had

a plume height reported by ground observers at about 10 km, is responsible for the enhanced frequency of Type 1 observations below 13 km around Julian day 190 when Type 1 frequencies reach 70% though the overall extinction





coefficient enhancement is low. All of the enhancements in the summer of 2001 are located in the Type 1 zone except, as shown in Figure 11c, a few Type 2 observations that appear in August 2001 near 14 km which, given the timing, may also be associated with the Bezymianny eruption.

*British Columbia Fire*

The data record for the SAGE III mission aboard the International Space Station began in June 2017 and was followed two months later by the British Columbia fire of August 2017 (Bourassa et al., 2019) which substantially surpassed all the smoke events observed by SAGE II in 20+ years of observations. Figure 12a shows the zonal mean 1020-nm extinction coefficient for northern mid-latitudes in 2017 from we infer a 1020-nm optical depth increase of

0.0037 or almost double the maximum optical depth enhancement observed by SAGE II (0.002). This enhancement is the largest observed by a SAGE instrument in the northern hemisphere. Extinction ratio, shown in Figure 7f and Figure 12b, appears to asymptote toward 2 with increasing extinction coefficient in a pattern that is similar to that seen in smaller events like Norman Wells. The figure shows that the fire-related aerosol is initially located primarily below 19 km but rises to at 23 km in the months following the event possibly due to diabatic self-lofting of aerosol

(Bourassa et al., 2019). In Figure 12c, the aerosol extinction coefficient enhancement for 2018 shows that the enhancement can be easily seen as late as mid-2018 or for about 1 year after the event. This is due to a combination of factors including the size of the enhancement in an otherwise fairly low aerosol period and due to the lofting leading to a longer residence time in the stratosphere compared to the smaller events observed by SAGE II. Essentially, all of the enhanced aerosol occurs as Type 1 and the frequency of these types, as shown for 2017 in

Figure 12d, reaches 50% of all observations below 22 km by Julian day 240 and 80% by Julian day 270 where it remains through the rest of the calendar year. The extent and longevity of the enhanced aerosol stands out compared to its SAGE II equivalents.

*Australian Fires of 2019/2020*

By far the largest fire-related perturbation to the stratosphere is associated with the widescale 2019/2020 Australian

brush fires. These fires began as early as September 2019 and persisted into February 2020 and produced large stratospheric impacts. For the purposes of the analysis of this event, we expand the usual time frame from a single year to two full years to facilitate understanding the scope of this event. Figure 13a shows the 10-day average 1020-nm extinction coefficient for 30 to 60S for 2019 and 2020. Despite significant fire activity in late 2019 and particularly in December 2019, there is little evidence of significant intrusions of smoke that pass TBA filtering until

early January 2020 (around Julian day 370 where day 1 is 1 January 2019) when a distinct layer around 17 km is noted. In the second half of that month into February a substantial enhancement is noted over a broad range of altitudes. SAGE III/ISS observations in through Julian day 420 show enhanced aerosol primarily located below 20 km but with some enhancement between 15 and 27 km in a distinct layer. While the highest extinctions in the densest parts of the layer begin their recover toward prior aerosol levels, there is clearly an enhancement in the mean

extinction coefficient above 20 km that eventually stretches as high as 33 km. The peak optical depth for this band is 0.011 or about 3 times the maximum optical depth enhancement from the British Columbia fire of 2017 and over 5



times larger than any non-volcanic enhancement seen in the SAGE II record. Overall, the enhancement in aerosol extinction coefficient remains clearly visible in this latitude band through the middle of 2021.

As with other events shown in the analysis, the aerosol extinction ratio tends to asymptote toward a fairly consistent value with increased extinction coefficient. In other events the asymptotic value has been on the order of 2 but with this event, as shown in Figure 13b the value is closer to 3.  This suggests some difference in the properties of the aerosol that comprise this event though, as previously discussed, it is not possible to infer exactly what those differences are beyond that they are likely some combinations of factors that modify the refractive index of the aerosol and/or the size of the aerosol. Figure 13c shows that essentially all of the aerosol enhancements for this event fall into the Type 1 category. Aerosol of this type become dominant below 15 km around Julian day 370 as over 80% of all observations. This becomes essentially all observations below 18 km through Julian day 600 but frequency decreases rapidly after that to just a few percent of all observations by the end of 2020. It is interesting to note that while the enhancement of aerosol extinction coefficient to such unusual altitudes is extremely interesting, they are, by the counting mechanism employed herein, identified in at most 1% of all events above 25 km and at most 10% for all events above 20 km.

*August Complex, California*

Despite its common name, the August Complex fires of 2020 are a series of fires in California that span from August through October of that year (Keeley and Syphard, 2021).  For this event, as shown in Figure 14a, enhanced aerosol is observed in the SAGE III/ISS data from Julian day 244 (September) through the end of the year that yields a maximum 1020-nm optical depth enhancement of about 0.001. Unlike other events where the maximum optical depth occurs over a narrow time frame, the maximum for the August Complex occurs in a broad period encompassing October through the end of the year. The extinction ratio, shown in Figure 14b, shows a distinct minimum below 15 km beginning about day 290 and extending to the end of the year. Like other events, at the largest extinction coefficient values despite low overall frequency of enhanced aerosol, the asymptotic value is close to 2 though the overall frequency of enhanced aerosol exceeds 1% (and never 5%) mostly between 12 and 16.5 km after about Julian day 250.

**5 The 1991 Baie-Comeau pyrocumulus and Mt. Pinatubo**

The eruption of Mt. Pinatubo (15N) in June 1991 is by far the largest stratospheric aerosol perturbation of the space-based measurement era (after 1978) increasing the stratospheric optical depth in the tropics to levels in excess of 0.2 (Kovilakam et al., 2020) and creating a global stratospheric aerosol enhancement that persisted to about 2000. This event had its first major eruption on 12 June (day 163) with others leading up to a main eruption on 16 June (day 167). Figure 15 shows the scatter plots, 1020-nm extinction coefficient versus extinction ratio, for 1991 for 30-day periods from Julian day 151 through Julian day 330. These plots are quite different than those for those fire events described above as they show a clear cluster of observations occurring in the Type 2 space that forms with a center near extinction coefficient 0.001 km$^{-1}$ and extinction ratio of 4 that grows in frequency through at least the 271-300



period with a similar extinction coefficient value but with slowly decreasing extinction ratio. This cluster was initially interpreted by Thomason (1992) as a new particle size mode created by the Mt. Pinatubo eruption. Later interpretations (Fromm et al., 2010; Gerisamov et al., 2019) noted the likely presence of smoke from the Baie-Comeau (49N) fire (24-30 June 1991; Julian days 175-181) and Fromm et al. (2012) suggested that the ubiquitous

observation of enhanced aerosol in the summer of 1991 was primarily associated with the fire event while observations from Tomsk, Russia (56N) (Gerasimov et al., 2019) suggest the presence of aerosol from Pinatubo in the lower stratosphere by mid-July. If the former assessment is correct, then the Baie-Comeau event would be a candidate for the largest smoke event in the SAGE record and thus a critical analysis of this pair of events is important.

Initially, the scatter of enhanced aerosol extinction coefficient in northern mid-latitudes (30-60N) 151-180 shows a mix of enhancements that are disperse across both the Type 1 and Type 2 regions. These data primarily occur in the last week of June so potentially appropriate for either source. However, the Type 2 events occur exclusively at latitudes below 40N and increase in frequency toward 30N in an altitude range between 12 and 18 km and are observed beginning on day 174.  The early Type 1 observations also occur primarily below 40N in the altitude range

between 13 and 19 km but do not appear consistently until Julian day 178. There is a gap in observations in this latitude band between Julian days 182 and 200.  By Julian days 201-210, both enhancement types are observed over a broad range of latitude primarily below 20 km. In the 211-240 Julian day period, when the total number of observations are relatively low, we observe essentially no Type 1 enhanced observations but an increased relative frequency of Type 2 enhancements which are shown in Figure 16a for all of 1991. Here we see the low frequency of

Type 2 events in late June becoming, by day 200, nearly all of the observations below 15 km with substantial presence up to 20 km. The high frequency of Type 2 observations persists until Julian day 300 when the combination of Type 1 enhancements at these altitudes and a loss of data due to the overall stratospheric opacity of the stratosphere due to the Mt. Pinatubo eruption effectively terminate Type 2 observations. Observations by SAGE II in the southern mid-latitudes are informative. Figure 17a shows the distribution of observations for Julian days

211-240 which, while not identical to northern hemisphere observations, is very similar to the northern hemisphere shown in Figure 15d (the same time period) with a mix of observations in both Type 1 and Type 2 areas. In figure 17b, the frequency of Type 2 observations is less extensive than in the northern hemisphere but still nearly ubiquitous below 15 km after Julian day 220. These must be associated with the Mt. Pinatubo eruption and cannot be related to the Baie-Comeau fire.

It is interesting that there are also Type 2 observations made above 25 km after the denser parts of the Mt. Pinatubo arrive in both hemispheres. This suggests that the Type 2 observations in this period could be related to the mechanism suggested by Thomason et al. (2020). This paper addresses the observed behavior of small to moderate eruptions and how extinction ratio is related to the extinction coefficient enhancement in the core of these eruptions. These events show a tendency to have high extinction ratio for smaller extinction coefficient enhancements thus

similar to Type 2 observations. It is plausible that less optically dense parts of a much larger event like Mt. Pinatubo's 1991 eruption could produce similar increases in extinction ratio. In fact, center of the Type 2 cluster in



Figure 15b has an extinction ratio of ~4 for an extinction coefficient enhancement of 0.001 km$^{-1}$ that is consistent with the relationship shown in Figure 8b of that publication. This process may come into play both in the lower altitudes where extinction coefficient levels are low but also high altitudes above the main aerosol layer where, as shown in Figure 16a, observations of Type 2 aerosol are inferred above optically dense portions of the aerosol layer as shown in 1020-nm extinction coefficient analysis for 1991 depicted in Figure 16b. Alternatively, as shown in Figure 8, aerosol extinction ratio in the range observed during this period is nominally possible with a brown carbon composition. However, that no other SAGE-observed smoke event produces a significant number of Type 2 observations much less such a unique feature. When seen in conjunction with the southern hemisphere observations, this makes the association of aerosol enhancements in the northern hemisphere in the summer of 1991 with the Baie-Comeau fire rather than the Mt. Pinatubo eruption untenable as one is forced to infer a completely different mechanism for the formation of these aerosol than any other pyrocumulus or fire-related event seen by a SAGE instrument from a fire event that is otherwise unremarkable. Type 1 events, shown in Figure 16c, occur at low frequencies below 22 km until about Julian day 240 when they become common between 17 and 25 km and dominate below 28 km by the end of the year as the Mt. Pinatubo aerosol spreads across the northern hemisphere.

We do not exclude the presence of smoke in the northern mid-latitudes during the summer of 1991. The work by Fromm et al (2010) and by Gerasimov et al. (2019) suggest that smoke is present in the lower stratosphere through at least July. For instance, Fromm et al. (2010) inferred, based on back trajectory analysis, for an extremely enhanced aerosol observation (>0.01 km$^{-1}$) that tracks back well to the time and location of the Baie-Comeau fire (Fromm et al., 2012). However, the extinction ratio for this observation is 1.04 so thus is a distinctly Type 1 observation and consistent with smoke observations elsewhere in the SAGE record. Similarly, smoke observations reported by Gerasimov et al. (2019) between 12 and 16 km are also observed in the SAGE II data set over a similar altitude range in late June and early July 1991 as depicted in Figure 16c. The extinction ratios of these SAGE II observations are between 1 and 3 and produce an average ratio of 2.6-2.8 when mixed with non-enhanced aerosol observations in the extinction ratio analysis shown in Figure 16a. These observations are also consistent with other observations of smoke by SAGE II. While these observations are consistent with smoke, we cannot conclude that all Type 1 observations in this period in this latitude band are smoke from the Baie-Comeau fire. SAGE instruments cannot reliably distinguish between aerosol like ash and smoke. As a result, attributing source of each of the Type 1 observations to either Baie-Comeau or large aerosol from one of the several Mt. Pinatubo eruptions cannot be made definitively. Ultimately, it is clear that the vast bulk of enhanced aerosol observed in the northern hemisphere in the second half of 1991 are associated with the Mt. Pinatubo eruption and not the Baie-Comeau fire and that Thomason (1992) is essentially correct in its inference of an interesting feature of volcanic impacts on the stratosphere. This feature can be viewed as a precursor for the work in Thomason et al. (2020). Given the presence of volcanic aerosol, inferring a magnitude for the enhancement of stratospheric optical depth by the fire component of this pair of events is not possible.





## 6 Conclusions

While we do not attempt to identify the source of isolated outliers in SAGE aerosol extinction coefficient measurements in the stratosphere, we nonetheless detect clusters of enhanced extinction coefficient from 13 smoke events in the SAGE II and SAGE III/ISS data sets. Of these, we are able to compute a column 1020-nm optical depth enhancement for 9 events that range from 0.0005 to 0.011. The remaining four events are a combination of low frequency of observations and somewhat hidden by the presence of other variability in stratospheric aerosol levels so that it is difficult to detect a meaningful column signal. Excluding 1992 through 1995 due to high aerosol levels from the Mt. Pinatubo eruption of 1991, the SAGE II period has 5 fire events from which a column optical depth perturbation may be determined over 17 years of observations or roughly 1 every 3 years. All but one of these events are in the northern hemisphere. The maximum optical depth values in the SAGE II period have several similarly sized events on the order of 0.002. These values are on the order of background stratospheric optical depth levels for mid and high latitudes. The SAGE III/ISS record has five total events of which we were able to compute optical depth values for 4. With a record approaching 6 years, this works out to about 1 measurable optical depth every 1.5 years or about twice the rate observed in the SAGE II record. Furthermore, while 2 of the SAGE III/ISS fire events are quite small, the other two, the British Columbia fire of 2017 and the Australian bushfires of 2019/2020 are the two largest events in the combined data set at roughly double (0.0037) and five times (0.011) the size of the largest events in the SAGE II record. Since both of these large events occur in the SAGE III/ISS record and in light of the rate difference between the two periods, it is tempting to infer a trend or, at least, a change between the two periods, however, the data records are relatively short compared to the observed rate of the fire events. It is also difficult to formulate a statistical test to compare the rate of fire events impacting the stratosphere between the two periods particularly given the low numbers of events. Another complicating factor is that the largest fire event in the record (the 2019/2020 Australian brushfires) is not even the largest known Australian brushfire in areal extent. For instance, the 1974/1975 Australian brushfire season, consumed over 117 million hectares (https://knowledge.aidr.org.au/resources/bushfire-new-south-wales-1974/) compared to 24.3 million hectares for the 2019/2020 season (Binskin et al., 2020). While it is likely that area-burned is not a good stand-in for stratospheric impact, the 1974/1975 season occurs prior to global space-based observations of stratospheric aerosol and its effect on the stratosphere is undocumented. At this point, it remains possible that the Australian brushfire season of 2019/2020 is simply a rare event rather than a harbinger of change and that using even relatively long records that the combined SAGE record provides to infer a change in smoke frequency should be undertaken with caution.

Using the SAGE instruments' 525 and 1020 nm aerosol extinction coefficient measurements to characterize these events yields some interesting results. In particular, we observe that the larger and more widespread events show a tendency for the ratio of these measurements to asymptote toward relatively well-behaved value for progressively larger extinction coefficient values. When this asymptotic value is observed, the value is usually close to 2 except for the large Australian brushfires of 2019/2020 where the value is closer to 3. This contrasts to observations of large aerosol such as ice clouds where the asymptotic value is about 1. This suggests that, while acknowledging substantial uncertainty in composition and size distribution of the observed smoke-based aerosol, the particles



dominating extinction must be relatively small probably smaller than 0.3 µm and perhaps show a surprising degree of consistency from event to event. An aerosol perturbation in the summer of 1991 has been attributed primarily to both Mt. Pinatubo and a fire in Baie-Comeau, Canada. While we infer the possible presence of smoke in limited numbers of SAGE II observations in June and early July 1991, we find that the distinct high extinction ratio cluster observed in this period is inconsistent with the optical behavior of all other smoke events observed by SAGE instruments and is simultaneously observed in the southern hemisphere strongly indicating that this feature of enhanced aerosol events in this period is the result of the Mt. Pinatubo eruption and not the Baie-Comeau fire.

The current version of GloSSAC (v2.1) underestimates the impact of pyrocumulus on stratospheric aerosol levels during the SAGE II period as the current cloud-clearing algorithm confuses smoke in the vicinity of the tropopause with cloud presence. This is not an issue for the SAGE III/ISS period where a more robust cloud algorithm (Kovilakam et al., 2023) can distinguish between cloud, smoke, and enhance volcanic aerosol. Porting this algorithm used for SAGE III to SAGE II should be straightforward and we expect substantial improvements to the SAGE II depictions, primarily the Norman Wells event, in the next release. Given the episodic nature of SAGE measurements at mid to high latitudes, the ability to observe an event in real time is a matter of some serendipity. This is demonstrated by the contrast of SAGE II and SAM II observations of the Daxing'anling Mountains fire in 1987 where the SAM II observations of that event are generally more ubiquitous and more intense than how it is observed in the SAGE II record. While the SAGE II and SAGE III/ISS records are unlikely to entirely miss a significant event as they will persist and spread across a greater latitude extent, nonetheless, the way that these events are observed in time and space may impact the interpretation of how intense they are and other characteristics of their optical properties.

The frequency of low-intensity volcanic impacts and pyrocumulus and other fire events on the stratosphere are sufficiently similar that some mixing of the event types is inevitable. This has been seen at least in the case of the Norman Wells fire event and the Korovin eruption in 1998 and possibly the Chisholm fire in 2001 with an eruption by Bezymianny. An event that we do not consider herein, an aerosol perturbation by the 2020 eruption of Raikoke, may have also been a mix of pyrocumulus affects and the volcanic aerosol (Ohneiser et al., 2019, Boone et al., 2022, Knepp et al., 2022). While GloSSAC does not discriminate aerosol modulations by their source or composition, other applications may be affected by unresolved composition knowledge and the possibility of diverse coincidence events should be considered. A corollary to the sampling issue are fire events like the August Complex of 2021 which persist for long periods of time that possibly inject aerosol into the stratosphere multiple times and turn what is normally a discrete event into a more protracted and perhaps less obvious enhancement.

We generally find that most smoke events remain or are observable only in the lowest most stratosphere and only the largest events are observed to have appreciable impacts above 20 km and even these do not persist for long periods of time. Their relatively short lifetime is somewhat ordained to follow that seen for mid and high latitude volcanic events which similarly do not persist for long periods due to the general circulation of the stratosphere. Only the two largest events remain readily observable in the SAGE III/ISS record for as long as a year with the British Columbia event lasting about 1-year as a distinct enhancement and the Australian brushfire effect lasting



perhaps 18 months. We do not suggest that the smoke-based aerosol is entirely gone but simply that the aerosol extinction levels have returned roughly to the values observed prior to the enhancement. It should also be noted that

given the very different optical properties of the smoke-based aerosol relative to sulfuric acid aerosol, particularly significant absorption at visible and near infrared wavelengths, that equivalent optical depth enhancements do not imply equivalent climate effects. Finally, it must be acknowledged that the impacts of pyrocumulus and other fire events on stratospheric are small and transient compared to the moderate to large volcanic events, particularly ones that occur in the tropics, that are also a part of the SAGE record. Chief among these is the Mt. Pinatubo eruption of

1991 which caused a global optical depth enhancement of about 0.2 at 1020 nm (Thomason et al., 2018) and persisted for nearly a decade. It remains, by far, the outstanding event affecting stratospheric aerosol levels since the start of space-based measurements in 1978.

**Code and data availability.**

SAGE II (https://doi.org/10.5067/ERBS/SAGEII/SOLAR_BINARY_L2-V7.0) and SAGE III/ISS data

(https://doi.org/10.5067/ISS/SAGEIII/SOLAR_HDF4_L2-V5.2) are accessible at the NASA Atmospheric Sciences Data Center. GloSSAC v2.1 (http://doi.org/10.5067/GLOSSAC-L3-V2.1) is available from the same location.

**Author contributions.**

LWT developed the analysis tools used throughout the paper and was the primary of author of the manuscript. TNK provide insight into the optical characteristics of smoke particles including Figure 8 and associated text.

**Competing interests.**

The authors declare that they have no conflict of interest.

**Financial support.**

LWT and TNK are supported by NASA's Earth Science Division as a part of the ongoing development, production, assessment, and analysis of SAGE data sets.






**Table 1.  Significant volcanic aerosol events in the stratospheric component of the SAGE II and SAGE III/ISS records**

| Date | Location |
|---:|:---|
| *November 1985* | Nevado del Ruiz, Colombia |
| *January 1990* | Kelut, Indonesia |
| *June 1991* | Mt. Pinatubo, Philippines |
| *August 1991* | Cerro Hudson, Argentina |
| *September 2002* | Ruang, Indonesia |
| *January 2005* | Manam, Indonesia |
| *April & July 2018* | Ambae, Vanuatu |
| *June & August 2019* | Ulawun, Papua New Guinea |
| *June 2019* | Raikoke, Russia |
| *March 2021* | La Soufriere, St. Vincent |
| *January 2022* | Hunga Tonga-Hunga Ha'apai, Tonga |




**Table 2.** **Table of smoke events and other events discussed in this paper. Smoke events are in bold with other event types denoted in the source column.**

| Date | Event source | Altitude range (km) | 1020-nm optical depth enhancement |
|---|---|---|---|
| *May 1987* | **Daxing'anling Mountains, Heilongjiang, China** | <14 | 0.002 |
| *September 1988* | **Yellowstone, Wyoming, US** | <11 | - |
| *February 1989* | PSC outbreak | 14.5-25 | 0.0005 |
| *July 1990* | **Circle, Alaska, US** | | 0.0012 |
| *June 1991* | **Baie-Comeau, Quebec, Canada**/Mt. Pinatubo Eruption, Philippines | <16 | - |
| *July 1998* | (Prob.) Korovin Eruption, Alaska, US | <13.5 | 0.001 |
| *August 1998* | **Norman Wells, Northwest Territories, Canada** | <19 | 0.002 |
| *April 1999* | Shishaldin Eruption, Alaska, US | 11.5-20 | 0.0008 |
| *February 2000* | Hekla Eruption, Iceland | <11 | 0.0005 |
| *May 2001* | **Chisholm, Alberta, Canada** | <17.5 | 0.0018 |
| *July 2001* | Bezymianny Eruption, Russia | <14 | - |
| *January 2003* | **Canberra, Australia** | <18 | 0.0005 |
| *August 2003* | **Conibear Lake, Alberta, Canada** | 16-17 | - |
| *August 2017* | **British Columbia, Canada** | <23 | 0.0037 |
| *August 2018* | **Redding, California, US** | <14 | - |
| *2019/2020* | **(Much of) Australia** | <30 | 0.011 |
| *August-October 2020* | **August Complex, California, US** | 11-18 | 0.001 |
| *June 2021* | **McKay Creek, British Columbia, Canada** | <15 | 0.0005 |




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





**Figure 1.** SAGE II 1020-nm aerosol extinction coefficient data at 11 km, above the reported MERRA tropopause altitude, and between 30 and 60N: a) all qualifying data, b) data remaining after GloSSAC-based cloud clearing, c) data limited to data 0.5 to 1.0 km above the reported tropopause, d) data limited to data 1.0 to 1.5 km above the reported tropopause, e) data limited to data 1.5 to 2.0 km above the reported tropopause, and e) data limited to data 0.5 to 1.0 km above the reported tropopause.





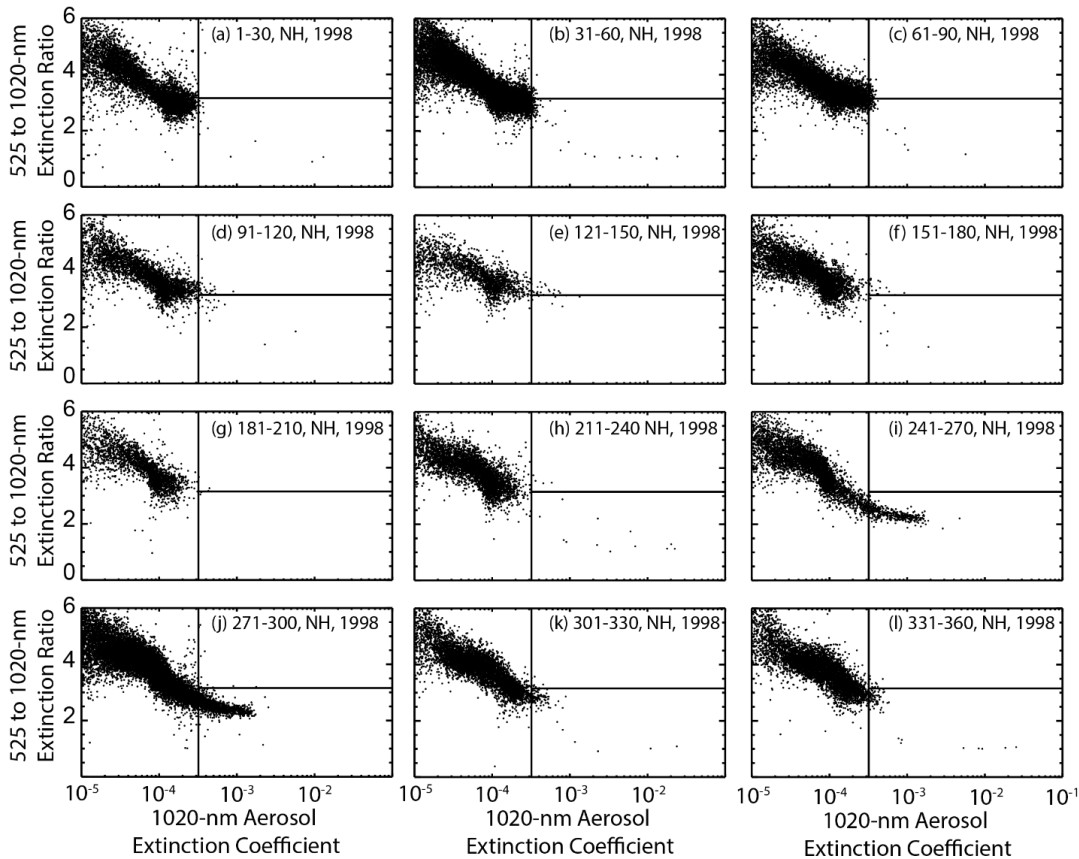

**Figure 2.** SAGE II 1020-nm aerosol extinction coefficient data plotted versus the 525 to 1020-nm extinction coefficient ratio for 1998 with all data above 1.5 km above the tropopause between 30 and 60N in 30-day increments from Julian 1-30 in frame a through 331-360 in frame l. The lines in each frame roughly divide the data into unenhanced, enhanced with a decreased extinction ratio, and enhanced with an increased extinction ratio using a technique described in the text.


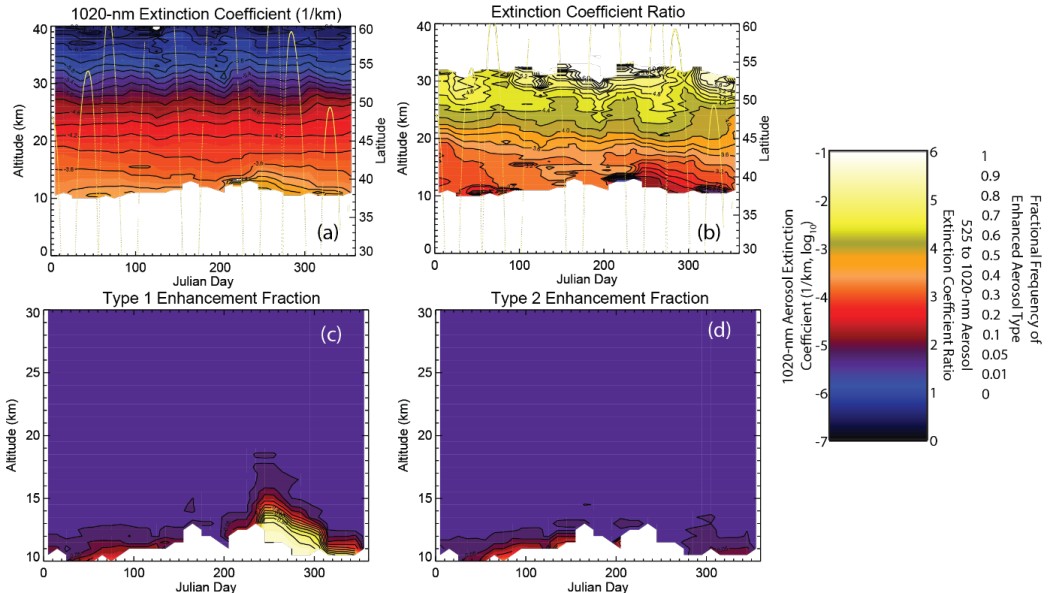

**Figure 3.** SAGE II aerosol data analysis for the 1998 Norman Wells event in the northern hemisphere (30-60N) in 10-day averages for 1020-nm extinction coefficient (a), extinction ratio (b), Type 1 enhancement fraction (c), and Type 2 enhancements fraction (d). Figures a and b also include the latitude of SAGE observations in yellow. Occurrences of the Type 1 and Type 2 observations follow from the discussion in the text. The color bar in this figure is applicable to all further color contour plot figures. Contours in extinction coefficient are spaced 0.2 in log-based 10 space. Extinction ratio contours are spaced in 0.2 increments and contours in Type frequency occur at 0.01, 0.05, and then every 0.1 from 0.1 to 1.0.



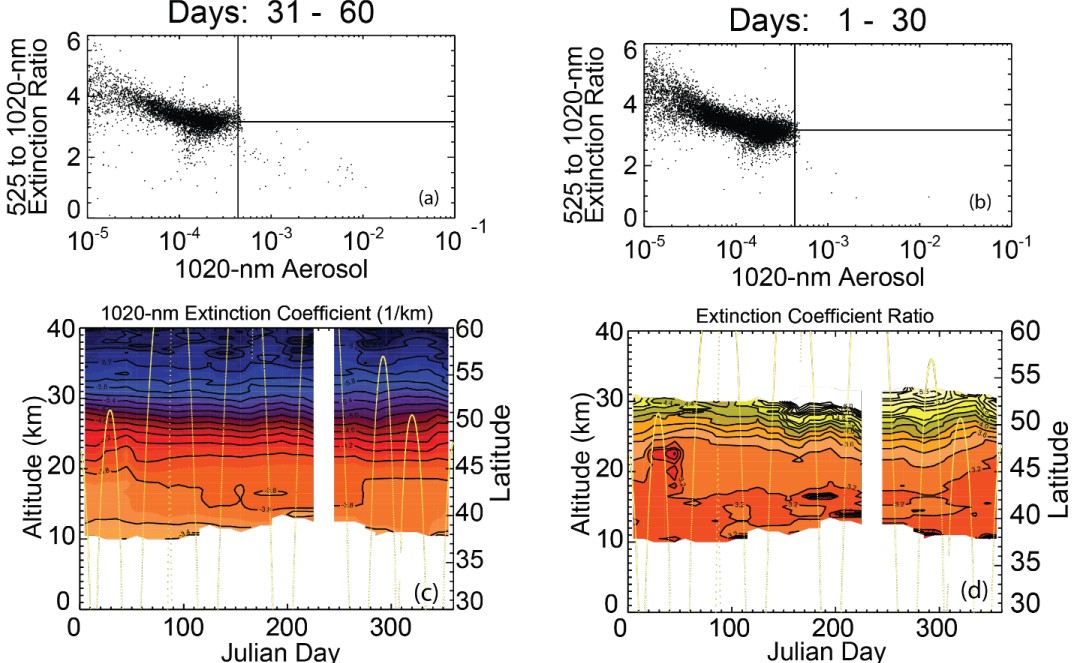

**Figure 4.** SAGE II 1020-nm aerosol extinction coefficient data plotted versus the 525 to 1020-nm extinction coefficient ratio for the 1989 PSC outbreak with all data above 1.5 km above the tropopause between 30 and 60N in 30-day increments for Julian days 31-60 (a) and 1-30 (b). Also, SAGE II aerosol data analysis for 1989 (30-60N) in 10-day averages for 1020-nm extinction coefficient (c) and extinction ratio (d). The color bar from Figure 1 applies to this figure. Contours in extinction coefficient are spaced 0.2 in log-based 10 space. Extinction ratio contours are spaced in 0.2 increments.



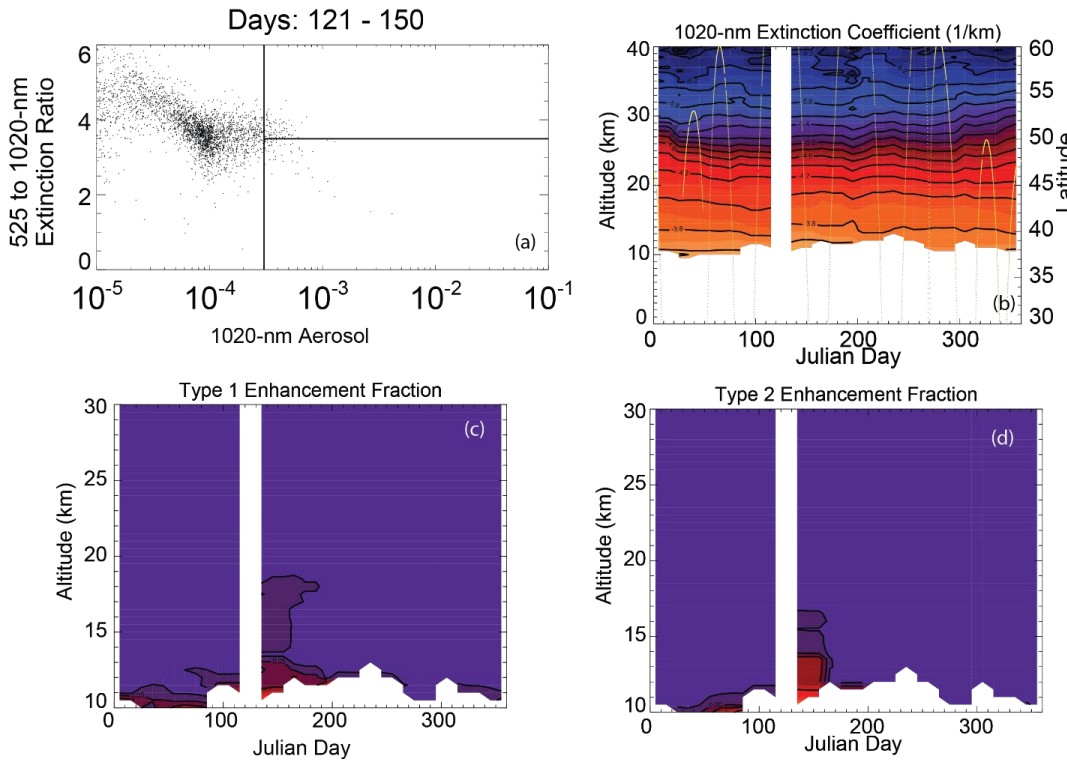

**Figure 5.** SAGE II 1020-nm aerosol extinction coefficient data plotted versus the 525 to 1020-nm extinction
coefficient ratio for the 1999 Shishaldin eruption with all data above 1.5 km above the tropopause between 30 and
60N Julian days 151-180 (a), SAGE II aerosol data analysis for 1999 (30-60N) in 10-day averages for 1020-nm
extinction coefficient (b), Type 1 enhancement fraction (c), and Type 2 enhancements fraction (d). Occurrences of
the Type 1 and Type 2 observations follow from the discussion in the text. The color bar in this figure is applicable
to all further color contour plot figures. Contours in extinction coefficient are spaced 0.2 in log-based 10 space.
Extinction ratio contours are spaced in 0.2 increments and contours in Type frequency occur at 0.01, 0.05, and then
every 0.1 from 0.1 to 1.0.

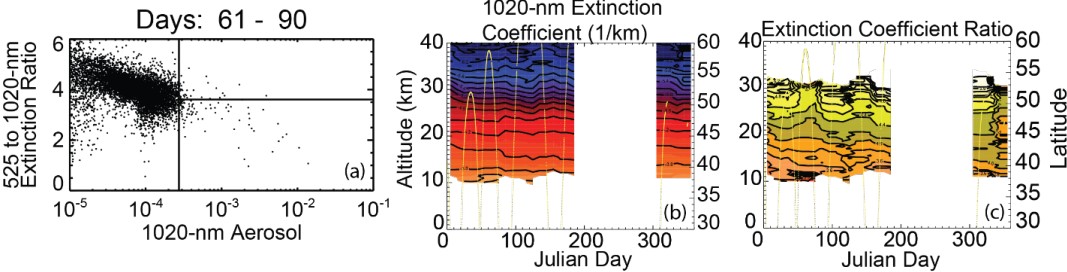


**Figure 6.** SAGE II 1020-nm aerosol extinction coefficient data plotted versus the 525 to 1020-nm extinction coefficient ratio for the 2000 Hekla eruption with all data above 1.5 km above the tropopause between 30 and 60N Julian days 61-90 (a), SAGE II aerosol data analysis for 2000 (30-60N) in 10-day averages for 1020-nm extinction coefficient (b), and aerosol extinction ratio (c). Contours in extinction coefficient are spaced 0.2 in log-based 10

space. Extinction ratio contours are spaced in 0.2 increments and contours in Type frequency occur at 0.01, 0.05, and then every 0.1 from 0.1 to 1.0. The color bar from Figure 3 applies to this figure.

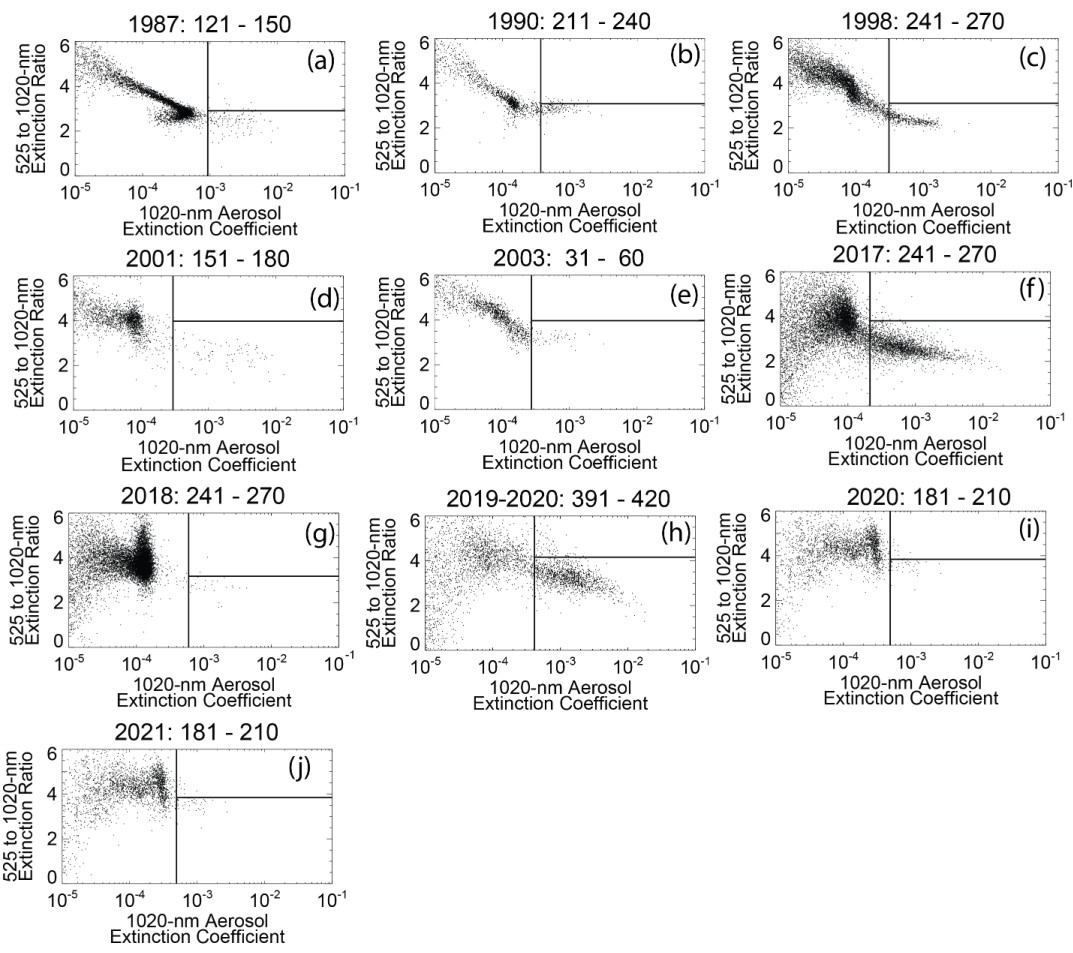


**Figure 7.** SAGE II 1020-nm aerosol extinction coefficient data plotted versus the 525 to 1020-nm extinction coefficient ratio for all the identified fire-related events in the SAGE II and SAGE III/ISS data sets including Daxing'anling Mountains China (a), Circle, Alaska, US (b), Norman Wells, Northwest Territories, Canada (c), Chisholm, Alberta, Canada (d), Canberra, Australia (e), British Columbia, Canada (f), Redding, California, US (g), Australia (h), August Complex, California, US (i), McKay Creek, British Columbia, Canada (j).




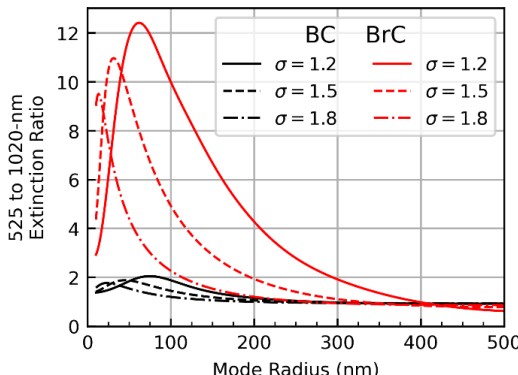

**Figure 8.** Theoretical extinction ratio values as a function of mode radius for single mode log-normal size distributions for black carbon (black) and brown carbon (red) with a range of widths and using complex refractive information from Bergstrom et al. (2002) for black carbon and Sumlin et al. (2018) for brown carbon.




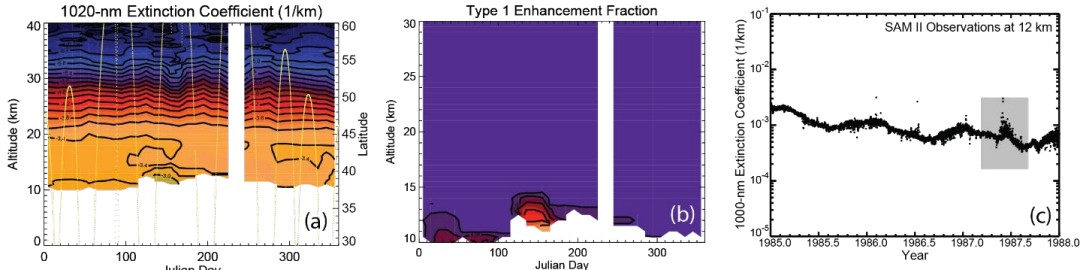

**Figure 9.** SAGE II aerosol data analysis (30-60N) for the 1989 Daxing'anling Mountains event in 10-day averages for 1020-nm extinction coefficient (a), the Type 1 frequency based on this data (b), and aerosol extinction coefficient data from SAM II at 12 km. The color bar from Figure 1 applies to this figure. Contours in extinction coefficient are spaced 0.2 in log-based 10 space and the frequency contours are at 0.01, 0.05, and then every 0.1 from 0.1 to 1.0.



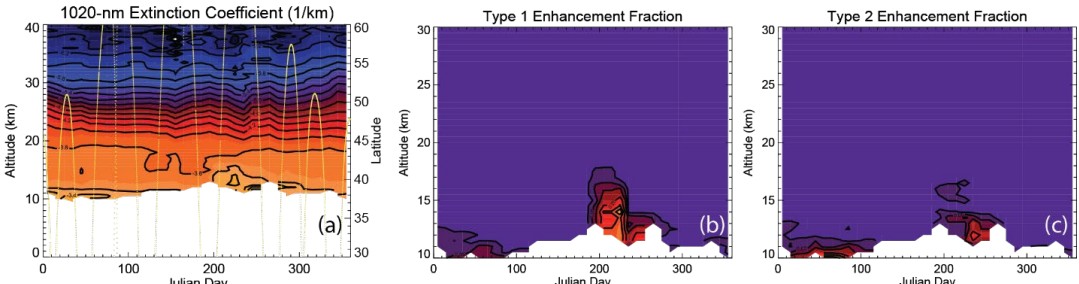

**Figure 10.** SAGE II aerosol data analysis (30-60N) for the 1990 Circle Alaska event in 10-day averages for 1020-nm extinction coefficient (a), the Type 1 frequency (b), and the Type 2 frequency (c). The color bar from Figure 1 applies to this figure. Contours in extinction coefficient are spaced 0.2 in log-based 10 space and the frequency contours are at 0.01, 0.05, and then every 0.1 from 0.1 to 1.0.






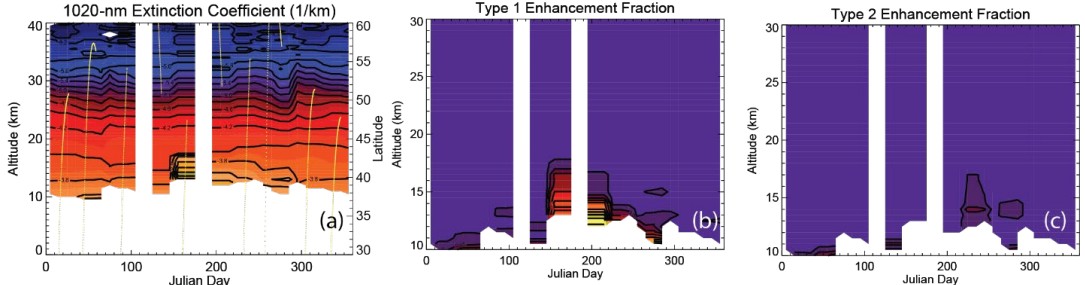

**Figure 11.** SAGE II aerosol data analysis (30-60N) for the 2001 Chisholm, Alberta, Canada event in 10-day averages for 1020-nm extinction coefficient (a), the Type 1 frequency (b), and the Type 2 frequency (c). The color bar from Figure 1 applies to this figure. Contours in extinction coefficient are spaced 0.2 in log-based 10 space and the frequency contours are at 0.01, 0.05, and then every 0.1 from 0.1 to 1.0.


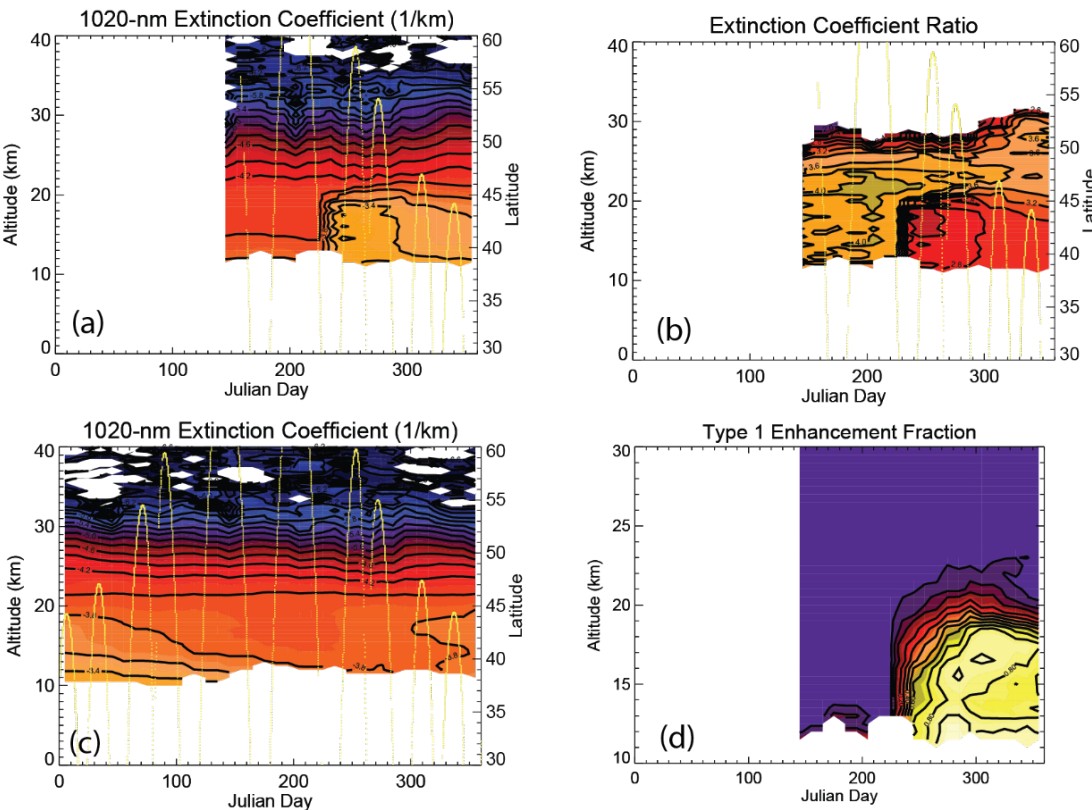

**Figure 12.** SAGE III/ISS 1020-nm aerosol data analysis for the British Columbia event of 2017 (30-60N) in 10-day averages for 1020-nm extinction coefficient (a) and aerosol extinction ratio (b). SAGE III/ISS 1020-nm aerosol data analysis in 10-day averages for 1020-nm extinction coefficient in 2018 (c), and Type 1 frequency in 2017. Contours in extinction coefficient are spaced 0.2 in log-based 10 space. Extinction ratio contours are spaced in 0.2 increments and contours in Type frequency occur at 0.01, 0.05, and then every 0.1 from 0.1 to 1.0. The color bar from Figure 3 applies to this figure.




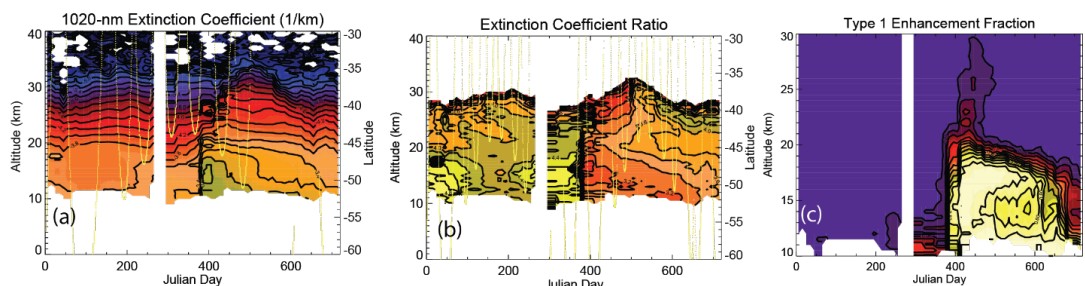

**Figure 13.** SAGE III/ISS 1020-nm aerosol data analysis for the Australian Fires of 2019/2020 (30-60S) in 10-day averages for 1020-nm extinction coefficient (a), aerosol extinction ratio (b), and Type 1 frequency Note that the analysis spans two calendar years. Contours in extinction coefficient are spaced 0.2 in log-based 10 space. Extinction ratio contours are spaced in 0.2 increments and contours in Type frequency occur at 0.01, 0.05, and then every 0.1 from 0.1 to 1.0. The color bar from Figure 3 applies to this figure.





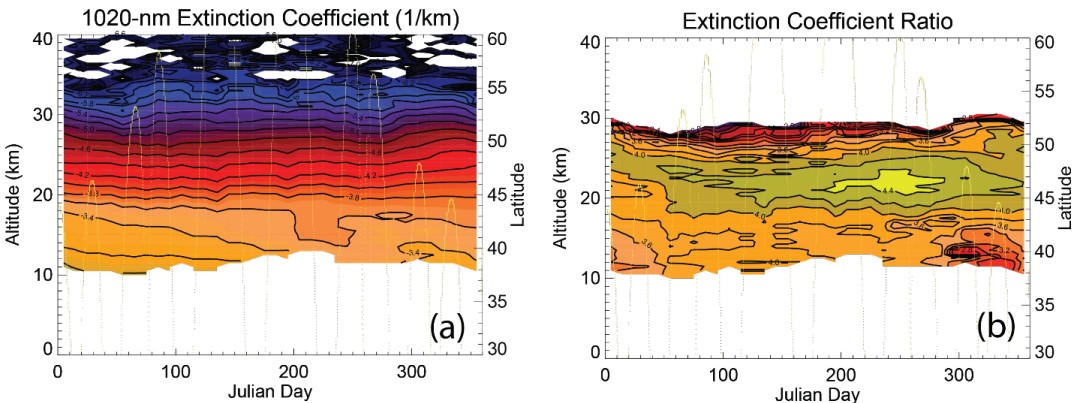

**Figure 14.** SAGE III/ISS 1020-nm aerosol data analysis for the 2020 August Complex in California (30-60N) in 10-day averages for 1020-nm extinction coefficient (a) and aerosol extinction ratio (b). Note that the analysis spans two calendar years. Contours in extinction coefficient are spaced 0.2 in log-based 10 space. Extinction ratio contours are spaced in 0.2 increments and contours in Type frequency occur at 0.01, 0.05, and then every 0.1 from 0.1 to 1.0. The color bar from Figure 3 applies to this figure.



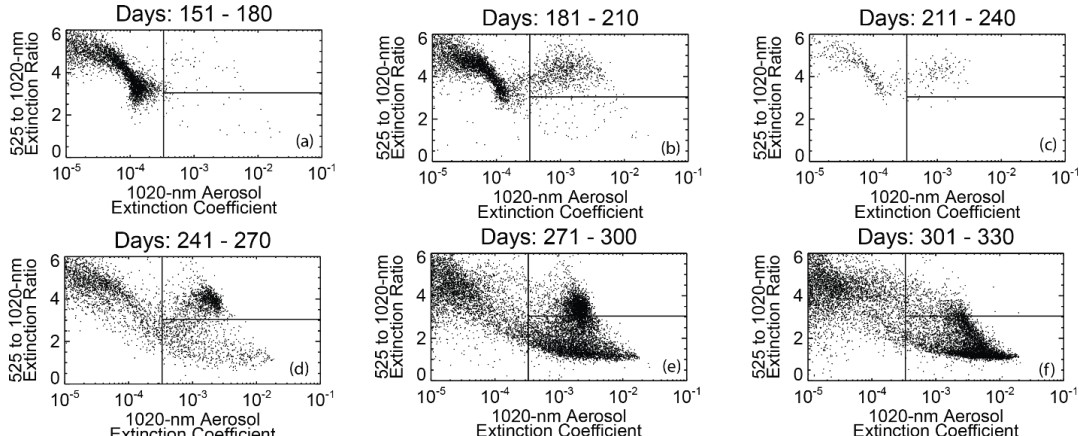

**Figure 15.** SAGE II 1020-nm aerosol extinction coefficient data plotted versus the 525 to 1020-nm extinction coefficient ratio for Julian day periods from 151-180 (~May) through 301-330 (~November) in the northern hemisphere in 1991.




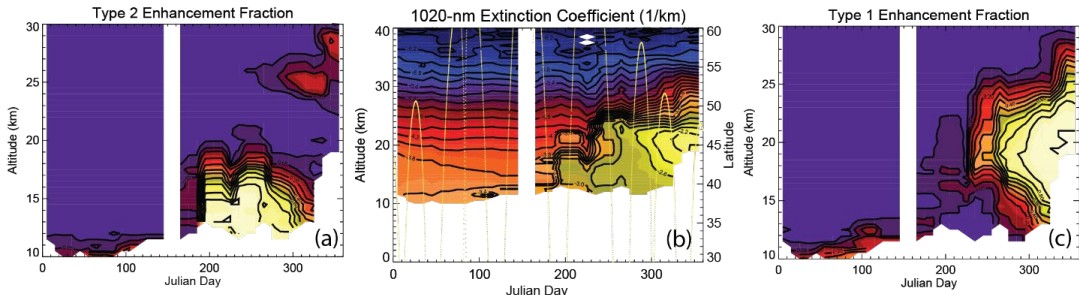

**Figure 16.** SAGE II 1020-nm aerosol data analysis for 1991 (30-60N) in 10-day averages for Type 2 frequency (a), 1020-nm extinction coefficient (b), and Type 1 frequency (c). Contours in extinction coefficient are spaced 0.2 in log-based 10 space. Contours in Type frequency occur at 0.01, 0.05, and then every 0.1 from 0.1 to 1.0. The color bar from Figure 3 applies to this figure.



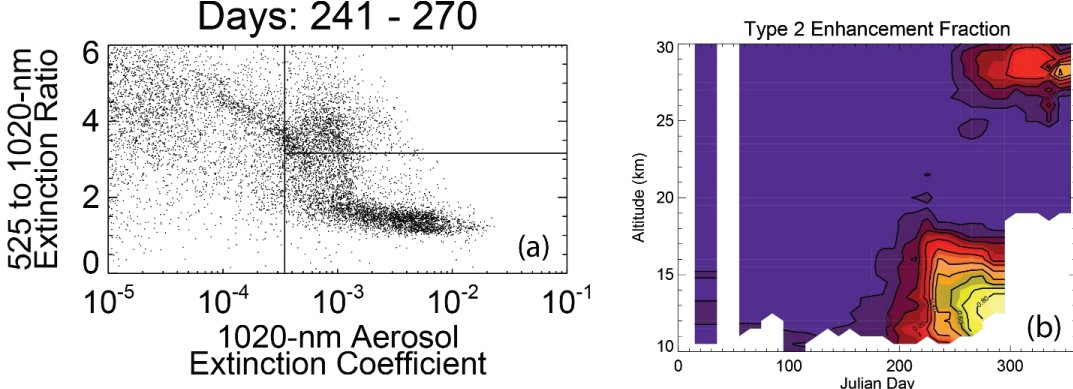

**Figure 17.** SAGE II 1020-nm aerosol extinction coefficient data plotted versus the 525 to 1020-nm extinction coefficient ratio for 1991 with all data above 1.5 km above the tropopause between 30 and 60S for Julian days 241-270 (a), and Type 2 enhancement fraction (c). Contours in Type frequency occur at 0.01, 0.05, and then every 0.1 from 0.1 to 1.0. The color bar from Figure 3 applies to this figure.