# Peer review of "Quantifying SAGE II (1984-2005) and SAGE III/ISS (2017-2022) observations of smoke in the stratosphere"

_EGUsphere, 2023_

## Author Response (AR1)

**Response to Reviewer 1:**

Perhaps not my finest moment in figure production for which I apologize. I have made substantial improvements to their readability and corrected missing or incomplete labeling and fixed the duplicate scatter plot in Figure 7i. I also cleaned up some of the associated captioning.

We have clarified in the text that when we discuss particle size particularly using a value that we are referring to particle radius (in several locations). We have also clarified that when we refer to the smoke aerosol as 'relatively small' we are doing so in comparison to the inferred size for other aerosol types (mostly ice clouds and large volcanic aerosol) that exhibit an asymptotic behavior in extinction ratio with increasing extinction coefficient in SAGE data (around lines 405 and 640).

The horizontal lines use in Figure 7 and a number of other figures are determined by observations of extinction coefficient and extinction ratio just prior to the event highlighted in individual frames. As a result, as aerosol properties vary with time, the line separating LRE and HRE zones also varies. The method for computing these lines is discussed in section 2 primarily in the third paragraph.

**Response to Reviewer 2:**

This comprehensive and pertinent study uses aerosol extinction coefficient derived from the SAGE II and SAGE III/ISS solar occultation measurements to identify and characterize smoke events in the stratosphere. A novel typing approach is presented that allows for discrimination of smoke from clouds and volcanic aerosol. The connections that are drawn between the asymptotic values of extinction ratio and the complicated space of particle composition and size distribution are important points of this paper. The combined Baie-Comeau and Mt. Pinatubo case is interesting and adds a convincing new element of analysis to these events, which have been studied in conjunction in previous works. Cautionary statements about inferring fire frequency from these data sets alone are appropriate. A few minor improvements are suggested below. The figures need attention as noted by the first reviewer.

*The figures and associated captions have been polished up significantly.*

Type 1 and Type 2 labelling is a bit confusing to follow. More intuitive labelling like Type LER (low extinction ratio) and Type HER (high extinction ratio) might be better. The authors could also consider a third type in the parameter space with very low values of extinction ratio, i.e. below ~2.

*We changed from Types to labels of LRE (low ratio enhancements) and HRE (high ratio enhancements).*

What does "not generally noted" refer to on line 176? In the literature?

*These minor events are unnoted completely or perhaps in a single publication. Since we are focused on relatively small events it seemed worthwhile to note their presence (and the outlier search found these for us). We have noted that they are not often noted in the literature as appearing in the SAGE data sets.*

Given the difficulties discussed on determining the enhancement values due to these events, how is this actually done?

*We integrate the column optical depth for the 10-day periods and essentially take the largest post-event value and subtract the lowest value immediately before the event for something like an impulse value. For larger events, this is pretty easy and fairly robust. As the event size gets small, how well we think it can be captured becomes less certain for the reasons described in the text. When our confidence in producing a meaningful value is too low, we choose not to report one. We've added some text around line 260 to indicate this.*

Sub-section 4.1 on non-smoke events along with figures 4,5,6 would be better placed at the end of section 4 to allow more immediate focus on the smoke events of interest.

*We have gone back and forth on this idea. In the end, we thought that these events provide some context of how minor non-smoke outlier events appear in the SAGE data sets particularly for phenomena that are well known from larger events and thus highlight how different these smoke events are from those events. To a lesser extent, moving them to the end would be anticlimactic in which case perhaps better to remove them altogether. Since for reasons mentioned above, we want to mention these events, we would prefer to leave the order as is.*

The extinction threshold line for type identification uses "the 99.5-percentile of 1020-nm extinction coefficient observations where it exceeds 10-4 km-1". However, based on Figure 7, this doesn't seem to be doing quite the right job for some of the events. Possibly the absolute 10-4 km-1 part of the criteria is over simplified?

*The main problem isn't with the $10^{-4}$ $km^{-1}$ value but that there is sometimes a notable trend in the extinction coefficient levels within the time frame of the analysis. Thus, when several months pass, the 99.5 cutoff becomes too large relative to the unenhanced aerosol at the time of the smoke (and other) events. This could nominally be fixed by pushing the calculation of closer to the event. However, this has a negative impact on the counting of LRE/HRE events such that very high fractions of such events are inferred when nothing is really going on other than on-going recovery from an earlier event (mostly volcanic). It was a bit of a trade off which we decided that the more conservative approach yielded the clearest outcomes. This was mentioned briefly near line 215. We have added an additional sentence about false positives in that section.*

The discussion on the "porting" of the algorithm to the GloSSAC cloud clearing in the conclusions is not clear. What is different about the SAGE II and SAGE III periods, and how is this not currently an issue for SAGE III?

*The SAGE II algorithm is fairly old (approach dates to version 6.2) and does not consider that smoke could occur in the stratosphere. As a result, as is shown in Figure 1, the algorithm confuses smoke with cloud presence. The new approach developed for SAGE III is painfully aware that smoke can be present in the stratosphere as well as volcanic material and ice clouds (at least at the tropopause). While the approach is not totally disjoint with the old technique, it does a better job parsing between the various types of aerosol that can be present in the stratosphere. We don't see a major impediment to using the new approach with SAGE II data, so we are hopeful for improvements in that part of the GloSSAC product in future releases. We have modified that discussion to hopefully make this clear.*